# Resilience of mobility network to dynamic population response across COVID-19 interventions: Evidences from Chile

**Pasquale Casaburi**[1,2], **Lorenzo Dall'Amico** [1], **Nicolò Gozzi**[1], **Kyriaki Kalimeri**[1], **Anna Sapienza** [1,3], **Rossano Schifanella**[1,4], **T. Di Matteo**[2,5,6], **Leo Ferres** [1,7], **Mattia Mazzoli** [1]*

**1** ISI Foundation, Turin, Italy, **2** Department of Mathematics, King's College London, London, United Kingdom, **3** Università del Piemonte Orientale, Alessandria, Italy, **4** Università degli Studi di Torino, Turin, Italy, **5** Complexity Science Hub Vienna, Vienna, Austria, **6** Museo Storico della Fisica e Centro Studi e Ricerche Enrico Fermi, Rome, Italy, **7** Universidad del Desarrollo, Santiago de Chile, Chile

* mattia.mazzoli@isi.it

**Data availability statement:** Data regarding socio-demographic variables at municipality

## Abstract

The COVID-19 pandemic highlighted the importance of non-traditional data sources, such as mobile phone data, to inform effective public health interventions and monitor adherence to such measures. Previous studies showed how socioeconomic characteristics shaped population response during restrictions and how repeated interventions eroded adherence over time. Less is known about how different population strata changed their response to repeated interventions and how this impacted the resulting mobility network. We study population response during the first and second infection waves of the COVID-19 pandemic in Chile and Spain. Via spatial lag and regression models, we investigate the adherence to mobility interventions at the municipality level in Chile, highlighting the significant role of wealth, labor structure, COVID-19 incidence, and network metrics characterizing business-as-usual municipality connectivity in shaping mobility changes during the two waves. We assess network structural similarities in the two periods by defining mobility hotspots and traveling probabilities in the two countries. As a proof of concept, we simulate and compare outcomes of an epidemic diffusion occurring in the two waves. While differences exist between factors associated with mobility reduction across waves in Chile, underscoring the dynamic nature of population response, our analysis reveals the resilience of the mobility network across the two waves. We test the robustness of our findings recovering similar results for Spain. Finally, epidemic modeling suggests that historical mobility data from past waves can be leveraged to inform future disease spatial invasion models in repeated interventions. This study highlights the value of historical mobile phone data for building pandemic preparedness and lessens the need for real-time data streams for risk assessment and outbreak response. Our work provides valuable insights into the complex interplay of factors driving mobility across repeated interventions, aiding in developing targeted mitigation strategies.

level in Chile are publicly available (https://www.ine.gob.cl/estadisticas/sociales/censos-de-poblacion-y-vivienda/censo-de-poblacion-y-vivienda). The information on mobility in Chile is extracted from mobile phone records based on antenna service. The data is proprietary. Interested researchers will be able to obtain access to the aggregated mobility flows used in this work in the same way as the authors did upon request to Alfredo Melo, Executive Secretary of the Ethics Committee - CEII of Universidad del Desarrollo (lmelo@udd.cl). Analysis of the anonymised mobile phone data was performed on the mobile operator's systems without transferring it outside. Only aggregated mobility patterns across municipalities were provided to researchers outside Chile, and only these have been used for the results presented here. Mobility data aggregated at municipality level from Spain are publicly available (https://www.transportes.gob.es/ministerio/proyectos-singulares/estudios-de-movilidad-con-big-data/estudios-de-movilidad-anteriores/covid-19/opendata-movilidad).

**Funding:** P.C., K.K., L.F., L.D., N.G., M.M. acknowledge support from the Lagrange Project of the ISI Foundation, funded by Fondazione CRT. This research was supported by FONDECYT Grant No. 1221315 to L.F.. L.D. further acknowledges support from Fondation Botnar (EPFL COVID-19 Real Time Epidemiology I-DAIR Pathfinder). M.M. acknowledges support from the Horizon Europe project VERDI (101045989) as well as from the EOSC project SIESTA (101131957). The funders had no role in study design, data collection and analysis, decision to publish, or preparation of the manuscript.

**Competing interests:** The authors have declared that no competing interests exist.

## Author summary

Population response to public health interventions during the Covid-19 pandemic exhibited strong heterogeneities associated to socio-economic factors, labour structure and demographics in multiple countries. As multiple waves of infections occurred, local authorities had to rely on repeated interventions, which eventually eroded population adherence to mobility restrictions, playing an important role in the evolution of local epidemics. Here we aim at explaining how municipalities in Chile responded to two different intervention periods in Chile in terms of their socio-demographic profiles, epidemiological data and centrality metrics computed in the pre-intervention mobility network. We find a significant association for network metrics in explaining mobility reductions in the two periods of intervention, and despite differences of response in terms of specific demographic profiles like age and gender, we show that the structure of the Chilean mobility network from the pre-intervention period remained almost unaltered across waves. From a public health perspective, our work has important implications for pandemic preparedness since it shows the advantage of re-using historic mobility data for informing epidemic models to respond to future health threats.

## Introduction

The COVID-19 pandemic has significantly impacted people's social and behavioral lifestyles worldwide [1–3]. Indeed, people's routines were affected by government-imposed restrictions aimed at mitigating the virus spread, as well as by individual decision-making processes [4–6]. In both cases, adherence to non-pharmaceutical interventions (NPIs) varied significantly across the population and it was shaped by several factors, including social, demographic, economic variables, and epidemiological conditions [5,7–9]. Historically, mobility data have been largely leveraged to inform spatial transmission models and perform importation risk assessment in such a complex framework [10–15]. In this context, large-scale mobility datasets became critical non-traditional tools routinely employed to measure and analyze the effects of non-pharmaceutical interventions (NPIs) aimed at curbing the spread of SARS-CoV-2 on individual behaviors [16–20].

This was possible thanks to the wide availability of data from tech giants and telecommunications companies, providing real-time insights into population activity potentially linked to the virus transmission. However, in the post-pandemic era, data accessibility is undermined by the end of Data for Good programs, hampering our capacity to leverage up-to-date data streams. As a result, the lack of data inevitably hinders our capacity to build sustainable tools for pandemic preparedness and to respond to new epidemics in a timely fashion [21]. Given their high value and ability to account for population-level heterogeneities in adherence to public health interventions, it is thus crucial to profit from the historical mobile phone data collected during the pandemic and to explore the possibility of re-using them for future analyses and modeling [22,23]. There is a critical need to better understand and integrate human behavioral change into our models to better inform future population-tailored strategies.

Previous works studied the resilience of mobility patterns following shocks, such as extreme weather events [24,25], epidemics [2,15,26–28] or both [29]. Recent findings highlighted how demographic differences were associated to loss of adherence to repeated interventions [30,31] and to delayed recovery of baseline mobility patterns [27] jointly with local GDP and population density [29], whereas some aspects of individual level visitation patterns were never recovered [2], with different spatial and temporal impacts on urban and rural

areas[28]. Recent insights proved that mobility network connectivity of US counties remained almost unaltered during the first COVID-19 wave[22]. However, little is known about how demographic-associated behavioral change due to repeated interventions may have impacted the resulting mobility network structure, and this effect must be quantified.

In this study, we harness large-scale mobile phone data from Chile and Spain, two countries with different social segregation located on two different continents, collected during the COVID-19 pandemic in a consistent fashion. First, we characterize the changes in the mobility response of Chilean municipalities to interventions issued during the first and second waves of COVID-19 by relying, on top of the variables already considered in the literature (such as wealth, labor structure, and demographics), on additional metrics derived from the baseline mobility network. Our results suggest consistency of most factors across waves despite the variability of a few others. We perform a network analysis to characterize the mobility network's resilience to interventions during the two COVID-19 waves in Chile and Spain. Relying on epidemic modeling, we explore the possibility of reusing historical mobility data to inform disease spatial invasion models of new epidemics.

Our findings represent a step towards enhancing pandemic preparedness by enabling the re-use of non-traditional data sources, like mobile phone data, for predicting population response in health emergencies. Our work sheds light on the dynamic nature of population behavioral change to repeated interventions [30–32], and contributes to the design of tailored public health policies in response to infectious diseases.

## Materials and methods

### Mobility data in Chile

On Wednesday, March 18, 2020, the Government of Chile announced the State of Alarm [33], issuing non-pharmaceutical interventions (NPIs) as school closures and mobility restrictions on a set of municipalities, followed by further measures in the next weeks. The country passed to a tiered system of NPIs in July 2020 [34], but a fast resurgence of COVID-19 cases led the Government to announce a tightening of mobility restrictions on all regions on April 1, 2021 [35]. For readability, we define the first two weeks of March 2020 as the *baseline* period, $b$, representing the "business as usual" mobility network. We let $f$ be the *first wave*, the four weeks following March 16, 2020, and $s$ to be the *second wave*, the four weeks following April 1, 2021. Our study focuses on the weekly average mobility flows in these periods.

Telefónica Chile provided mobility data for Chile in the form of eXtended Detail Records (XDRs). This dataset records the starting time of a data-packet exchange session between a device belonging to an anonymized user and geolocated cell phone towers. The dataset covers the period from March 1, 2020, to April 28, 2021. To minimize noise in the data due to spurious stops not representative of a destination, e.g., devices stopping for a few minutes due to traffic, we defined *stays* as devices connecting to the same tower for at least 30 minutes. We filtered out data points not complying with this condition and obtained a dataset representing users' *stays*. We assigned cell towers to *comunas* (Chilean municipalities) using their coordinates, and we counted as *trips* all devices switching to a new tower placed at least 500 meters away. We discarded trips between two towers placed within the same municipalities; here, we only focused on external mobility.

### Mobility data in Spain

A national mobile phone operator collected mobility data for Spain, treated [36] and published by the Ministry of Transport, Mobility and Urban Agenda of Spain, *MITMA* in a

public and online repository [37]. The data describe the daily movements of individuals between Spanish municipalities from February 21, 2020, to March 18, 2021. Municipalities are mapped into a coarser spatial division in which small rural municipalities with low population density are grouped to include areas not covered by antennas [36]. This data collection is based on individuals' active events, e.g., users' calls, together with passive events, in which the user's device position is registered due to changes in the cell tower of connection. Similarly to the data treatment we performed for Chile, these trips were aggregated using users' movements between consecutive *stays* of at least 20 minutes in the same area, disregarding trips of less than 500 meters [36]. Here, we focus on external mobility. Hence, we discarded all records regarding trips within the same municipality, i.e., the diagonal of the OD matrices.

## Sociodemographic, epidemiological and mobility network metrics for Chile

Chile is characterized by a high level of disparity in socio-economical traits, a high variability of climate conditions from North to South, and a strong urban-rural divide, with the Metropolitan area of Santiago representing the most densely populated area of the country. This is reflected in high heterogeneities of wealth, educational level, median age, active working population, and labor structure across Chilean comunas. In the context of the COVID-19 pandemic, a complex interplay between the above spatial heterogeneities and the epidemic characterized the population response to interventions, with the demographic strata of the population behaving differently. To tackle these differences, we focused on the inter-municipal mobility post-interventions in two separate periods in Chile. Finally, we analyse the resulting mobility network of Chile during the two post-intervention periods and test the validity of our findings in an analogous case study in Spain. We refer the reader to S1 Text for the Spanish case study results and details on the resilience metrics employed.

Specifically for Chile, we collected most of the socio-demographic variables defined at municipality level from the Chilean 'Instituto Nacional de Estadistica' (INE) [38], while the municipal development index (IDC) was provided by the Universidad Autónoma de Chile (UAC) [39]. As an illustrative example, among the variables obtained from the INE, the continuous variable *urbanization* defined as the percentage of the population living in urbanized areas, allows us to analyze the extent to which Chile's urban-rural divide shaped population responses to interventions. The IDC is a composite index that encodes information on the development and welfare of each municipality. Epidemiological data, i.e., confirmed cases, active cases, deaths, and PCR tests by municipality of residence, were collected and made available by the Chilean Ministry of Science [40]. We extracted two variables, namely COVID-19 confirmed cases and PCR tests performed at municipality level, and aggregated these records at a weekly level to minimize noise due to reporting delays and weekends. To better represent the epidemic phases at municipality level, we averaged these data over the four weeks of the two waves periods. We extracted *deaths* records from the database of the *Department of Statistics and Health Information* (DEIS) and the Ministry of Health [41], encoding the number of COVID-19 related deaths in each municipality. See Table 1 for a comprehensive list of all variables collected.

To compare mobility between the first and second wave $\delta^2 M$ defined in Eq 3, we defined *case increment* as the relative increment of the variable *new cases* between the second and first wave, instead of the incidence of cases.

**Table 1. All variables collected.** List of all variables collected in our dataset by description, type, and source. The list includes the socio-economic, epidemiological, and network metrics extracted from the mobility data. *: variable was discarded post VIF test. Sources legends refer to [38] for INE, [39] for UAC and [40] for MinCiencia.

| Variable | Description | Type | Source |
|---|---|---|---|
| log(pop) | logarithm base 10 of the population | static | INE |
| pop density | population per square meters | static | INE |
| age | median of the population age | static | INE |
| urbanization | percentage of population living in urbanized areas | static | INE |
| gender ratio | number of males divided by the number of females | static | INE |
| schooling | median of schooling years of the population | static | INE |
| primary | share of workers employed in the first sector | static | INE |
| secondary* | share of workers employed in the second sector | static | INE |
| tertiary | share of workers employed in the tertiary sector | static | INE |
| dependency | ratio of non-employable over employable population | static | INE |
| employed | share of employed population (above 15 years old) | static | INE |
| IDC | municipal development index | static | UAC |
| new cases | weekly total incidence of COVID-19 new reported cases | dynamic | MinCiencia |
| active cases* | weekly average incidence of COVID-19 active cases | dynamic | MinCiencia |
| new deaths | weekly total mortality of COVID-19 | dynamic | DEIS |
| test rate | weekly average of PCR tests per 1000 inhabitants | dynamic | MinCiencia |
| out-strength pc | $S_{out}$, baseline out-strength per capita (outbound trips pc) | static | extracted |
| in-strength pc* | $S_{in}$, baseline in-strength per capita (inbound trips pc) | static | extracted |
| out-path-length* | $\langle l_{out}\rangle$, baseline outbound-path-length (peripherality) | static | extracted |
| in-path-length | $\langle l_{in}\rangle$, baseline inbound-path-length (peripherality) | static | extracted |
| clustering | $c$, baseline clustering coeff. (neighbors interdependence) | static | extracted |
| betweenness | $bc$, baseline betweenness centrality (key for connectivity) | static | extracted |

## Mobility data analysis

For both countries, we extracted weekly (and daily) origin-destination (OD) matrices $M_{ij,w}$ ($M_{ij,d}$) encoding the total number of trips between municipalities $i$ and $j$ that occurred in week $w$ on day $d$. These matrices define a time-dependent weighted directed network where nodes are municipalities, links are mobility routes, and link weights are the number of trips occurred in the considered time interval. The weekly aggregated OD matrices will be the main object of study of our work, whereas the daily OD matrices will only serve as a sensitivity test on the time scale of aggregation.

We defined the weekly averaged flows for the three periods as follows:

$$\begin{cases} \bar{M}_{ij,b} = \frac{1}{2} \sum_{w \in b} M_{ij,w} \\ \bar{M}_{ij,f} = \frac{1}{4} \sum_{w \in f} M_{ij,w} \\ \bar{M}_{ij,s} = \frac{1}{4} \sum_{w \in s} M_{ij,w} \end{cases} \tag{1}$$

We computed for each municipality $i$ the relative outbound mobility drop in the first and second waves with respect to the baseline as:

$$\Delta M_{i,f} = \frac{\sum_{j \neq i} \bar{M}_{ij,f} - \sum_{j \neq i} \bar{M}_{ij,b}}{\sum_{j \neq i} \bar{M}_{ij,b}} \quad \Delta M_{i,s} = \frac{\sum_{j \neq i} \bar{M}_{ij,s} - \sum_{j \neq i} \bar{M}_{ij,b}}{\sum_{j \neq i} \bar{M}_{ij,b}} \tag{2}$$

Negative values of $\Delta M_{i,f}$ reflect reductions in mobility from the baseline.

Analogously, we defined $\delta^2 M_i$ for each municipality $i$ as the relative change of the total outgoing mobility from $i$ observed during the second wave with respect to the first wave:

$$\delta^2 M_i = \frac{\sum_{j \neq i} \bar{M}_{ij,s} - \sum_{j \neq i} \bar{M}_{ij,f}}{\sum_{j \neq i} \bar{M}_{ij,f}} \tag{3}$$

Negative values represent lower outgoing flows from municipality $i$ during the second wave with respect to the first wave. To account for municipalities centrality and interdependence in the baseline period ("business as usual") network encoded by $\bar{M}_{ij,b}$, we defined standard network metrics, see Table 1 (we refer the reader to S1 Text Section *Network metrics* for their mathematical definition). These metrics, computed on the baseline mobility network, intrinsically capture physical connectivity and infrastructural limitations, which influence typical traffic patterns under a business-as-usual context. Infrastructural constraints are thus implicitly reflected in the baseline mobility data and are encoded in the network metrics employed as covariates.

## Regression model

Spatial lag and linear regression models are performed exclusively on the Chilean case study. First, we measured the Moran index [42] of the mobility drop $\Delta M_{i,P}$ of each Chilean municipality $i$ in Chile in the two periods $P \in \{f, s\}$, first and second wave respectively, to measure the influence of neighboring areas on the mobility change of Chilean *comunas*. We defined spatial proximity of municipalities using a binary Fuzzy contiguity matrix [43,44] $\mathbf{W}$, such that $W_{ij} = 1$ if and only if *comunas* $i$ and $j$ are neighbours. We found a Moran index of mobility change in the first wave period of $I_{M_f} = 0.36$ ($p_v < 0.001$) and in the second wave period of $I_{M_s} = 0.22$ ($p_v < 0.001$). Hence, we employed a Spatial Lag model [45] to explain each municipality $i$ outbound mobility drop $\Delta M_{i,P}$ (Eq 2) from the baseline period in the first and second waves, $P \in (f, s)$ respectively. The model takes the general form:

$$\mathbf{\Delta M_P} = \rho \mathbf{W} \mathbf{\Delta M_P} + \beta \mathbf{X_P} + \epsilon \tag{4}$$

where $\mathbf{X_P}$ is the covariates matrix. The model covariates are the baseline mobility network metrics (*e.g.* strength, clustering coefficient, betweenness centrality), sociodemographic variables (*e.g.* population density, gender and age distribution, urbanization level, labor structure), and COVID-related variables (*e.g.* incidence of new cases). As part of the epidemiological variables, the second wave model features two additional covariates, namely the number of deaths and the test rate, that were unavailable in the first wave. We refer the reader to Table 1 for a detailed description of the model variables. Differently from a linear model, an additional term weighted by $\mathbf{W} \mathbf{\Delta M_P}$ takes into account possible spatial autocorrelations, where $\mathbf{W}$ is the spatial proximity matrix defined above. The magnitude of the regression coefficients $\rho, \beta$ and their statistical significance are computed using a maximum likelihood estimator

[44–46]. A statistically significant coefficient $\rho$ means that spatial effects are present and the response of municipalities to NPIs is also influenced by neighbouring areas.

To compare the two different waves, we implemented a linear regression in which the dependent variable is now the relative change of the total outgoing mobility observed during the second wave with respect to the first wave, $\delta^2 M$ defined in Eq 3:

$$\delta^2 \mathbf{M} = \beta \mathbf{X} + \epsilon \tag{5}$$

Where $\mathbf{X}$ is again the covariates matrix. In this case the magnitude and statistical significance of the regression coefficients have been computed using an ordinary least squares estimator.

We standardized the covariates and performed a variance inflation factor test (VIF) [47] to avoid multicollinearity among the covariates. We excluded from all regressions four variables, namely *active cases*, *out-path-length*, *in-strenght pc*, and *secondary* as their VIF scored over 10.

## Spatial invasion model

We define a spatial invasion model on the networks on the first and second wave period, namely $\bar{M}_{ij,f}$, $\bar{M}_{ij,s}$. We simulate an epidemic invasion starting in the municipalities of Santiago Airport in Pudahuel and register the arrival times at municipality level for the next 28 days. We run $n_s = 500$ simulations and compute the average arrival time for each *comuna*, restricting to those who were invaded at least 20 times in all simulations, to compute averages on a minimal sample for each location. The model is a simplified SI (Susceptible-Infected) model in which municipalities can have two states at each time step, $I_m \in \{0, 1\}$, $I_m = 1$ for invaded municipalities and $I_m = 0$ for susceptible ones, we do not account for the internal transmission dynamic. We defined the force of invasion from a municipality $i$ on susceptible locations $j$ as:

$$\lambda_{ij} = \beta \, I_i \left(1 - I_j\right) \left(p_{ij} + p_{ji}\right) \tag{6}$$

where $\beta$ defines the probability of infection per contact, $I_i = 1$ and $I_j = 0$ in order to allow for invasion, and $p_{ij}$ and $p_{ji}$ are the traveling probabilities accounting for the mixing of $i$ and $j$ *comunas*. In the remainder of the paper, we set $\beta = 0.5$. Additional details on the definition of the traveling probabilities $p_{ij,P}$, are provided in S1 Text Section *Traveling probabilities definition*, together with an alternative definition in S1 Text Section *Alternative parametrization of the invasion model*.

## Results

### Mobility response to interventions in Chile

We focused on Chile's first and second wave of COVID-19 infections (see Methods for periods definitions). The two intervention periods differed significantly in terms of the number of restricted municipalities and the outbound mobility flows, as shown in Fig 1. While the first month of intervention in the first wave only involved a few *comunas* and had a significant impact on the outbound mobility, the first month of interventions in the second wave had a lower impact on the intensity of outbound mobility flows, despite involving the vast majority of Chilean *comunas*. We hypothesize that municipalities with different demographic profiles, cases incidence, and centrality in the baseline mobility network responded differently to interventions. We investigate the association of socioeconomic, epidemiological, and pre-intervention network metrics with mobility change with respect to the baseline via spatial lag

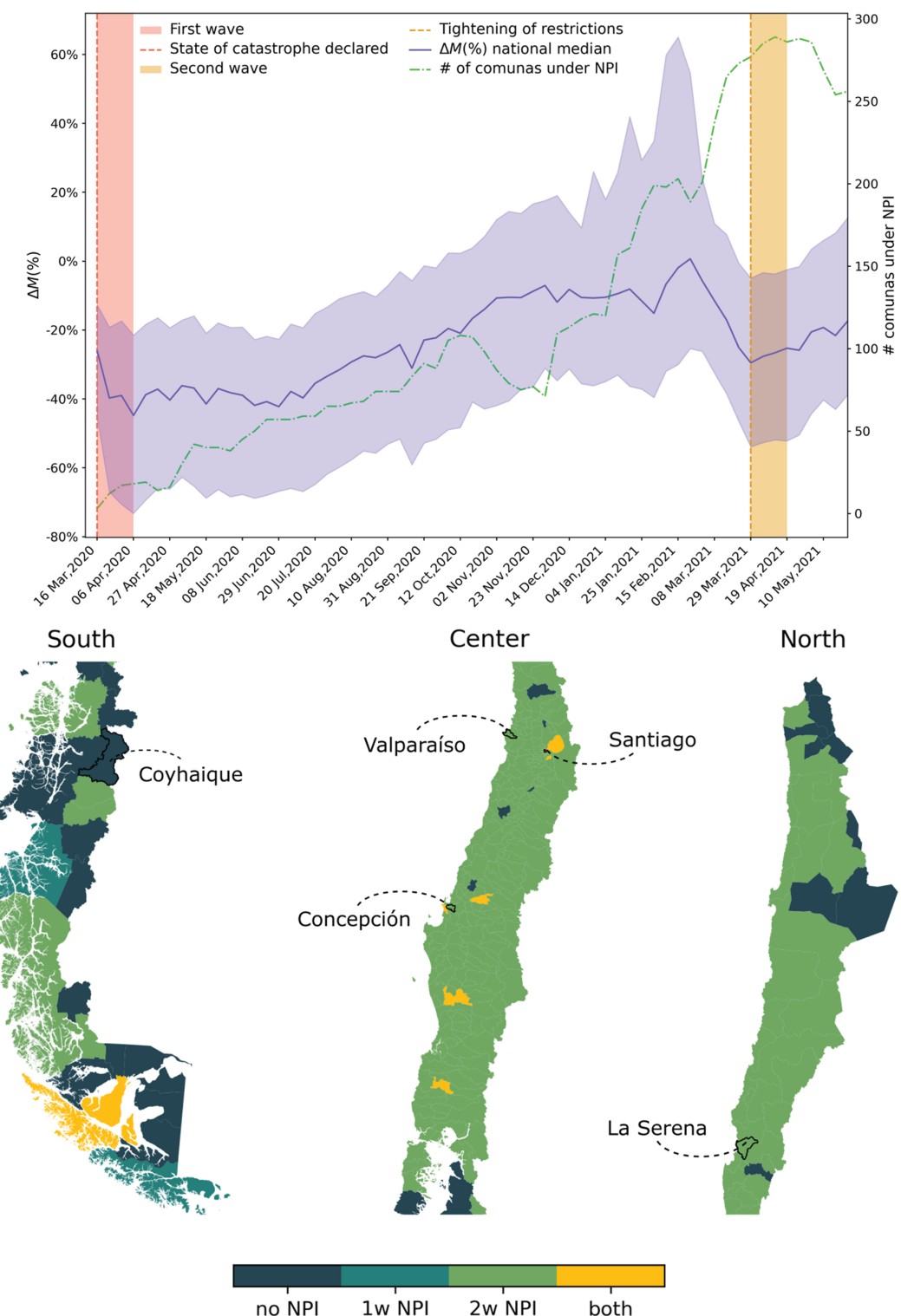

**Fig 1. The first and second wave in Chile.** On top: weekly median mobility change with respect to the baseline (purple solid line), and respective 95% interval across municipalities (purple shaded area). The dashed-dotted green curve represents the number of municipalities under restrictions, while the vertical areas correspond to the first (red) and second (yellow) waves. On the bottom, the map of three macro-regions of Chile shows municipalities under restrictions. The color code represents municipalities according to the NPIs they experienced. We make the distinction for adoption only in the first wave (light blue, 1w NPI), only in the second wave (green 2w NPI), in both waves (yellow, both) and neither (dark blue, no NPI). Administrative boundary data were obtained from BCN (https://www.bcn.cl/siit/mapas_vectoriales/index_html).

models to account for spatial autocorrelations. Most interestingly, we want to assess whether the response to the second period of interventions was associated with the same factors or if some ceased to explain population response. To do so, we employ a linear regression model to explain the two periods mobility differences.

**First and second COVID-19 waves.** The spatial lag model achieves high predictive performance for the first wave in terms of $R^2 = 0.53$ and Kendall Tau, $\tau_K = 0.53$. The detailed results are reported in the left column of Table 2. For some covariates – *e.g.* wealth (encoded as IDC) and labor structure – we recover the results already observed in the literature [7,9], showing that wealthier municipalities with higher employment in the tertiary sector achieved stronger mobility reductions than others. On top of this, our results enlist additional variables associated with mobility changes:

- the mobility reduction correlates *positively* with COVID-19 incidence and with the baseline municipalities betweenness centrality;
- the mobility reduction correlates *negatively* with the baseline neighbors' interdependence (*clustering coefficient*), the outbound trips per-capita (*out-strenght pc*) and average peripherality (*path-length*);
- the spatial autocorrelation coefficient is a statistically significant determinant, revealing the importance of municipalities neighbours influence.

**Table 2. Correlates of mobility change across the two waves in Chile.** *Tables in the first and second columns: first and second wave.* The spatial lag model coefficients, $\beta$ for covariates and $\rho$ for spatial lag from Eq (4) are reported with their 95% confidence interval and the goodness of fit measured by $R^2$ and the Kendall's Tau ($\tau_K$) coefficient between observed and predicted mobility drops. *Table in the third column: comparison between waves.* Linear regression coefficients $\beta$ and respective 95% confidence interval for the model of Eq (5). For all models, the goodness of fit is reported at the bottom of the table, and we report significance as: ***=99.9%, ** = 99%, * = 95%. *Cases increment* is employed only in the linear regression in place of *new cases*, on the rightmost table. Positive coefficients represent higher mobility flows associated with higher covariates.

| | First wave | | | Second wave | | | Waves difference | | |
|---|---|---|---|---|---|---|---|---|---|
| | **Coeff.** | [0.025 | 0.975] | **Coeff.** | [0.025 | 0.975] | **Coeff.** | [0.025 | 0.975] |
| intercept | -0.24*** | -0.29 | -0.21 | -0.21*** | -0.24 | -0.18 | -0.0 | -0.08 | 0.08 |
| age | 0.01* | 0.0 | 0.03 | 0.00 | -0.02 | 0.02 | -0.21*** | -0.32 | -0.1 |
| urbanization | 0.04*** | 0.03 | 0.07 | 0.06*** | 0.04 | 0.09 | -0.17* | -0.32 | -0.01 |
| gender ratio | -0.01 | -0.04 | 0.0 | 0.00 | -0.02 | 0.03 | 0.24** | 0.07 | 0.4 |
| dependency | -0.04*** | -0.06 | -0.02 | -0.07*** | -0.1 | -0.05 | -0.05 | -0.22 | 0.12 |
| schooling | -0.01 | -0.04 | 0.01 | 0.00 | -0.02 | 0.04 | 0.12 | -0.07 | 0.31 |
| primary | 0.00 | -0.02 | 0.02 | -0.03* | -0.07 | -0.01 | -0.31** | -0.5 | -0.12 |
| tertiary | -0.08*** | -0.11 | -0.06 | -0.11*** | -0.15 | -0.08 | 0.24* | 0.02 | 0.46 |
| pop density | 0.01 | -0.01 | 0.03 | 0.02* | 0.0 | 0.05 | 0.12 | -0.02 | 0.25 |
| log(pop) | 0.01 | -0.01 | 0.03 | -0.01 | -0.05 | 0.01 | -0.53*** | -0.72 | -0.35 |
| IDC | -0.03** | -0.06 | -0.01 | -0.04* | -0.08 | -0.01 | 0.31** | 0.1 | 0.52 |
| new cases | -0.01** | -0.03 | -0.0 | -0.00 | -0.02 | 0.01 | | | |
| clustering | 0.02** | 0.01 | 0.05 | -0.00 | -0.03 | 0.02 | -0.4*** | -0.56 | -0.24 |
| betweenness | -0.01** | -0.03 | -0.0 | -0.01 | -0.03 | 0.0 | 0.06 | -0.04 | 0.16 |
| in-path-length | 0.04*** | 0.03 | 0.05 | 0.10*** | 0.09 | 0.12 | 0.18*** | 0.08 | 0.27 |
| out-strength pc | 0.03*** | 0.02 | 0.05 | 0.01* | 0.0 | 0.04 | -0.25*** | -0.35 | -0.15 |
| $\rho$ (spatial lag) | 0.35*** | 0.24 | 0.46 | 0.19*** | 0.09 | 0.3 | | | |
| deaths | | | | 0.00 | -0.01 | 0.01 | | | |
| test rate | | | | 0.00 | -0.01 | 0.02 | | | |
| cases increment | | | | | | | -0.01 | -0.1 | 0.08 |
| | Pseudo $R^2$ = 0.53 | | | Pseudo $R^2$ = 0.58 | | | Adjusted $R^2$ = 0.49 | | |
| | $\tau_K = 0.53$*** | | | $\tau_K = 0.38$*** | | | $\tau_K = 0.38$*** | | |

The second wave period model suggests consistent correlates with respect to the first wave model, namely labour structure, urbanization, municipality development, baseline period mobility network metrics, i.e., peripherality (*inbound path-length*), outbound trips per capita (*out-strength*) and spatial autocorrelation. However, there are some differences between the two waves periods correlates: epidemiological variables, clustering and betweenness centrality are not statistically significant in the second wave, see the second column in Table 2.

**Differences across waves.**   We aim to define the factors associated with the altered population response in the second wave with respect to the first wave, hence describing behavioral changes in response to repeated interventions. To approach this question, we employed a linear regression to explain the relative change of outgoing mobility observed during the second wave with respect to the first wave $\delta^2 M_i$ (see Methods). Here, we replaced the variable *new cases* with *cases increment* to account for the relative change of cases incidence across waves.

The model achieves an adjusted $R^2$ = 0.49 and a statistically significant Kendall's Tau coefficient of $\tau_k$ = 0.38.

Our findings evidence that, with respect to the first wave, in the second wave period:

- higher mobility is observed in the municipalities with higher development index (*IDC*) and more active population employed in the tertiary sector;
- lower mobility is associated with higher urbanization indices , higher population, higher percentage of women, and higher age profiles.

Interestingly, we do not observe a significant role played by the increment of cases incidence.

In S1 Text Section *Sensitivity to NPI dummy variables*, we performed a sensitivity test on the inclusion of NPI covariates in the spatial lag models, finding that correlates significance and performance of the model are substantially unaltered (see Tables A and B in S1 Text).

## Network resilience

The analysis carried on in the previous section shows a wide overlap of factors associated with mobility reductions and a significant role played by network metrics computed on the baseline mobility network. These quantities are static in time and refer to the configuration of the pre-interventions mobility network. Their significance thus suggests a remarkable resilience of the mobility network structure that we investigated by focusing on the change of mobility hotspots and origin-destination flows across the two waves periods.

**Hotspots and traveling probabilities analyses.**   To compare the node features of the two networks, i.e. first and second wave mobility networks, we defined as hotspots the municipalities with the highest outbound mobility flows in each of the three periods using the Loubar method [48]. We identified hotspots by considering the total outbound mobility of each municipality, as this measure plays a critical role in shaping the structure of the mobility network. The total outbound mobility directly affects the traffic load on network links, influencing the network overall structure and its resilience in time. Details on the Loubar method are provided in S1 Text Section *Hotspots definition* and an alternative hotspots definition accounting for mobility per-capita is provided in S1 Text Section *Alternative definition of mobility hotspots*). In each period, we defined three levels of hotspots ranked by their (decreasing) mobility outflows and subdivided the municipalities into three sets. For each level, we adopted the Jaccard similarity index *J* to compare the overlapping hotspots across the three periods. The results are summarized in Fig 2.

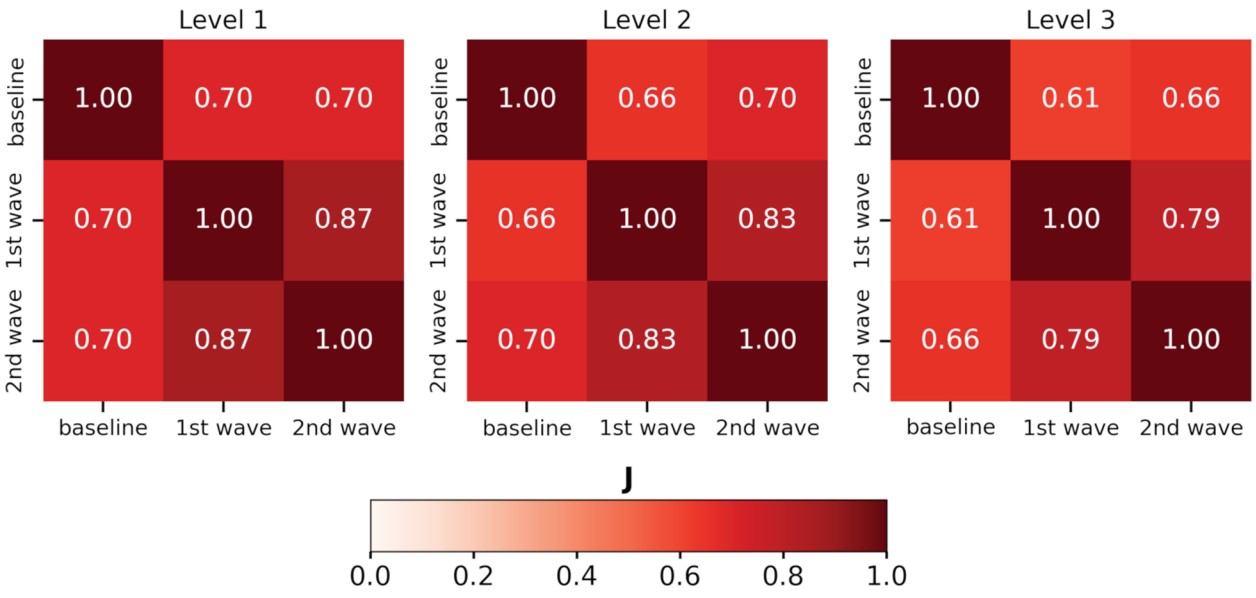

**Fig 2. Resilience of mobility hotspots across epidemic waves in Chile.** Heatmaps showing the Jaccard similarity *J* for three hotspot levels across the three periods, i.e., baseline, first, and second wave. Level increases from left to right, with Level 1 encoding *comunas* with the highest mobility flows.

The results evidence a large overlap at all three levels for all three periods. Interestingly, the hotspot configuration appears to be significantly stable between the first and the second wave, suggesting a significant correlation in the network structure.

To compare the link features of the two networks, we defined the probabilities of traveling from any municipality $i$ to $j$ as $p_{ij,P} = \frac{\bar{M}_{ij,P}}{\sum_{k \neq i} \bar{M}_{ik,P}}$, where $P \in \{b, f, s\}$, i.e. baseline, first or second wave period. Fig 3A and 3B show that the highest probability routes, on the top right of the plots, are highly correlated across the three periods, and hence, the probability of traveling over the major routes out of municipalities in Chile is highly similar across periods. On the other hand, minor routes, laying on the bottom left corner of plots, are characterized by lower flows and show high differences across waves.

A more in-depth analysis, highlighting additional resilient features of the mobility network over time, is provided in S1 Text Section *Further insights on the network features associated with network resilience*.

**Spatial invasion modeling.** As a proof of concept, we run an epidemic model to simulate the spatial invasion of an infectious disease arriving at Santiago's airport on both the first and the second wave mobility networks. The model is a susceptible-infected (SI) model, where locations can only get infected once, and the probability of invasion depends on the mobility flows between areas (see S1 Text Section *Traveling probabilities definition*). We register the average arrival time of the disease at the municipality level obtained by running $n_s = 500$ simulations and show the comparison for the two waves mobility networks. In Fig 3C, we show how structural similarities in the two epidemic waves mobility networks do not affect sensibly the outcomes of the predicted arrival times at municipalities. In Fig 3C, we observe a high correlation between arrival times of the first 14 days of simulations, which reflects the good correlation of the highest traveling probabilities $p_{ij}$ in Fig 3A and 3B. On the other hand, the uncorrelated arrival times in the third week of simulations, in the top right corner of Fig 3C, reflect the higher degree of variability between minor routes in the first and second waves, i.e.

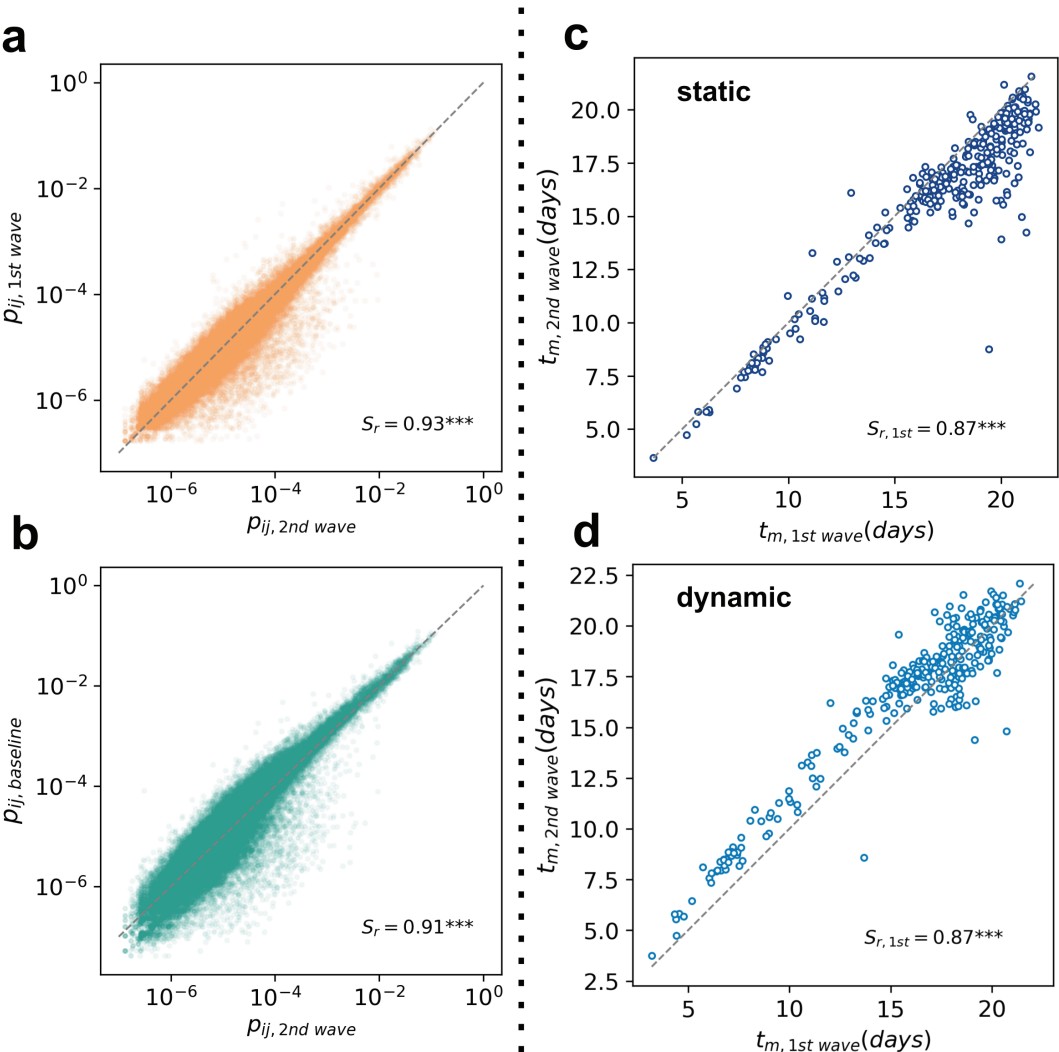

**Fig 3. Epidemic modeling on Chile's two waves mobility networks. (a)** Traveling probabilities $p_{ij}$ computed from the first wave network against those computed on the second wave network. **(b)** Traveling probabilities $p_{ij}$ computed from the second wave network against those computed on the baseline network. The grey dashed line is the identity diagonal. The quantity $S_r$ denotes the Spearman's correlation coefficient computed on traveling probabilities. **(c)** Average arrival times $t_m$ at municipalities resulted from the epidemic model run on the first and second wave static networks describing the average mobility flows observed in each period, and **(d)** on the dynamic networks. The dashed grey line is the line $x = y$, and $S_r$ is Spearman's correlation coefficient computed between the two modeled average arrival times.

the lowest traveling probabilities $p_{ij}$ from any *comuna i*, shown in Fig 3A and 3B. This result highlights how the main routes of mobility between comunas were preserved well across the two waves, while the highest level of variability was registered in the secondary routes, representing a lower share of outbound travels from municipalities. As a sensitivity test, we performed the same simulation over a dynamic version of the mobility networks, in which the two waves networks account for the daily flows between municipalities (see S1 Text Section *Traveling probabilities definition*). In Fig 3D, we show how results prove robust for temporal aggregation. In the S1 Text Section *Alternative parametrization of the invasion model*, we

tested an alternative parameterization of mobility data into the epidemic model that best accounts for mobility reduction scenarios, proving the robustness of our results.

## Validity of results in further countries

To check for the validity of our findings regarding the resilience of the mobility network across repeated interventions during the Covid-19 pandemic, we repeated the same methodology and analyses on the Spanish dataset and recovered consistent results for the Spanish case study in the three periods analogously defined (see S1 Text Section *Spanish network resilience*). In Fig A in S1 Text, we found a high overlap of mobility hotspots across the three periods, specifically with all Jaccard indexes above 60%, while in Fig B in S1 Text, we found a high correlation of traveling probabilities from Spanish municipalities, with Spearman correlations above $S_r = 0.89$ between first and second waves networks and $S_r = 0.92$ between second wave and baseline networks. Note that Spain counts with a higher number of municipalities with respect to Chile, 8132 vs 346 respectively, meaning that results are robust to different spatial configurations of the mobility network.

## Discussion

We focused on the municipality-level mobility response to COVID-19 first and second-wave interventions in Chile, where we observed consistent factors associated with post-intervention mobility reduction across the two periods. These factors included network metrics encoding mobility patterns from the baseline pre-interventions period. This provides evidence that both mobility networks resulting from two separate interventions in two periods share similar structural aspects. We looked into the structural features of the two periods' mobility networks and found that, despite changes in the behavioral response to interventions associated with specific demographic profiles, the two networks exhibited strong structural similarities. We characterized the similarity between the mobility networks of the first and second COVID-19 waves in Chile in terms of mobility hotspots overlap to account for node features and traveling probabilities correlations to account for the link features of the network. We expanded our analysis to the Spanish case study. By leveraging an open mobility dataset provided by the Spanish Ministry of Transportation [36,37], we reproduced our analyses on the Spanish case study in the three periods analogously defined (baseline, first, and second wave). Our approach proved robust since we observed strong structural similarities between the mobility network of the second and first waves in Spain.

This resilience suggests that certain areas and routes are critical to explaining mobility despite varying restrictions and changes in public behavior. This has strong implications for epidemic modeling: since the major mobility routes are preserved across waves, they encode preferential pathways of spatial invasion, as proved by the essentially unchanged disease arrival times predicted by the model on the two periods networks. This finding adds to previous knowledge on the impact of mobility network heterogeneity on the predictability of disease spread at international level [49], in which major air travel routes determine a higher probability of spatial invasion. Observing a resilient mobility network at sub-national level, hence preserving major traveling probabilities across interventions, yields high predictability of spatial invasion patterns within the country beyond the initial epidemic phase. This observation can be exploited to design *a priori* sentinel locations for epidemic surveillance of new strains, which represent the most at-risk of invasion locations in the country.

The spatial lag models reveal several key factors associated with mobility reduction during the first and second waves. Our analysis recovers some known relations already observed

in the literature, such as the role of the labour structure [7], the urbanization level, active population, and local incidence [9].

Unlike previous studies, our modeling approach accounts for network metrics of municipality centrality in the baseline mobility network, which play a significant role in the population response to interventions in both waves. The positive correlation between mobility reductions and betweenness centrality suggests that the most central municipalities were also the most responsive to interventions. This is relevant because these municipalities are likely the most important in the long-range disease spread, hence playing as bottlenecks for invasion patterns. Lower mobility reductions are instead associated with higher clustering – hence pertaining to groups of densely connected interdependent municipalities –, higher inbound path length – hence peripheral *comunas* –, and with higher amount of trips per capita. This behavior can be attributed to peripheral locations with a higher economic dependency on neighboring municipalities in granting essential services to the population. During the second wave, besides the still determinant socio-economic variables, the only network significant correlates were the inbound path length and the per capita out-strength: peripheral municipalities with typically higher trips per capita had lower mobility reductions than others.

Accounting for the effect of NPIs explicitly in the regression models required arbitrary choices on their parameterization. The two waves were characterized by two distinct restrictions, a binary system in the first wave and a tiered system in the second. Moreover, they are characterized by a very low variance in both waves as shown in Fig 1. A sensitivity test on the inclusion of NPI variables revealed the robustness of covariates significance (see S1 Text Section *Sensitivity to NPI dummy variables*).

Our analyses further show the key differences in several variables when explaining the mobility variations between the first and second waves. Factors like age, gender, and population did not impact the post-intervention mobility change with respect to the baseline, but they were associated with a change in response in the second wave with respect to the first. Scarcely populated rural and peripheral areas, with a higher development index that are more populated by men of younger age exhibited higher mobility levels in the second wave with respect to the first wave. This indicates that dynamic population response to repeated interventions may be more strongly associated with certain demographic profiles rather than being common to the overall population.

Finally, the study finds significant spatial autocorrelation in mobility responses. This suggests that municipalities did not act or react independently to interventions but were influenced by neighboring areas. This observation underscores the importance of considering spatial dependencies in pandemic response strategies.

Our work shows that, despite population behavioral change to repeated interventions in areas with specific demographic profiles, centrality metrics, and clustering in the pre-intervention mobility network, urbanization, development, and labour structure, the overall mobility network exhibits relatively high resilience to shocks driven by NPIs. From a public health perspective, this resilience can be exploited to improve surveillance and inform interventions with historical mobility data, predicting areas at the highest invasion risk and allowing efficient allocation of resources to anticipate local outbreaks. These findings have strong implications for pandemic preparedness since, in this context, data readiness can be built without depending on real-time data streams.

## Limitations

In this study, we did not consider local incidence stratified by demographic traits, which could have altered the population response of specific groups in the two waves.

While the mobility network we studied for the three periods, i.e. baseline, first, and second waves, is the result of an average over the first month post-interventions weekly flows, mobility is a dynamic process that can exhibit local, both spatially and temporally, fluctuations due to local holidays, seasonality, weather events, and regional climate. We chose to average weekly flows of four weeks post-interventions to minimize the effect of any of these factors on our analyses. In Fig 3D, we prove how our results are robust to temporal aggregation of mobility flows, i.e., considering the original dynamic network of daily mobility flows.

Our study did not address whether critical mobility routes identified in our analyses were critical or not to explain the observed epidemic outcomes during the COVID-19 pandemic in Chile. Specifically, our analysis did not address whether the routes identified as most critical were indeed the ones most affected by the pandemic or whether the municipalities connected by these routes experienced the greatest epidemic impact. Answering such questions is challenging with the data employed in this study, and more reliable data sources would be needed to provide a robust and accurate assessment. Additional details and explanations of these limitations, along with a preliminary analysis that could be further explored in future works, are provided in the S1 Text Section *Further epidemic impact insights*.

This study did not consider variables related to Chilean climate regions. Further research is needed to account for climate impact on the resulting population response to NPIs.

## Author contributions

**Conceptualization:** Lorenzo Dall'Amico, Nicolò Gozzi, Mattia Mazzoli.

**Data curation:** Pasquale Casaburi, Leo Ferres.

**Formal analysis:** Pasquale Casaburi, Leo Ferres, Mattia Mazzoli.

**Investigation:** Pasquale Casaburi, Leo Ferres, Mattia Mazzoli.

**Methodology:** Pasquale Casaburi, Lorenzo Dall'Amico, Nicolò Gozzi, Kyriaki Kalimeri, Anna Sapienza, Rossano Schifanella, Tiziana Di Matteo, Leo Ferres, Mattia Mazzoli.

**Project administration:** Mattia Mazzoli.

**Supervision:** Mattia Mazzoli, Tiziana di Matteo.

**Visualization:** Pasquale Casaburi, Mattia Mazzoli.

**Writing – original draft:** Pasquale Casaburi, Nicolò Gozzi, Kyriaki Kalimeri, Mattia Mazzoli.

**Writing – review & editing:** Pasquale Casaburi, Lorenzo Dall'Amico, Nicolò Gozzi, Kyriaki Kalimeri, Anna Sapienza, Rossano Schifanella, Tiziana Di Matteo, Leo Ferres, Mattia Mazzoli.

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
