## [Decision Letter · Decision Letter 0]

29 Sep 2024

Dear Mr. Mazzoli,

Thank you very much for submitting your manuscript "Resilience of mobility network to dynamic population response across COVID-19 interventions: evidences from Chile" for consideration at PLOS Computational Biology.

As with all papers reviewed by the journal, your manuscript was reviewed by members of the editorial board and by several independent reviewers. In light of the reviews (below this email), we would like to invite the resubmission of a significantly-revised version that takes into account the reviewers' comments.

I agree with the reviewers that this is an interesting paper demonstrating the importance of high-resolution mobility for measuring behavioral change during epidemics. I also agree with the reviewers that there are some aspects of the work which should be clarified and potentially some additional analysis are needed to support the conclusions (mostly depending on the resulting clarifications). For example, it's not clear how interventions were included in the regression model. Are they exclusively captured by in the mobility data and stratification of the analysis between intervention periods? If so, I might suggest a unified analysis with indicator variables associated with interventions (or continuous/ordinal). Second, the authors state that the network structures were strikingly similar. While I agree there are strong similarities, looking at the scatter plots of the Pijs there are many which have changed by 2+ orders of magnitude. I agree with the reviewers that more work is needed to articulate which aspects of the networks were retained and which were changed. I think you have this information in the regression analysis and do discuss it a bit, but I believe the work would be far strengthened with a deeper dive into differences/similarities in the networks. Third, I agree with the reviewers that contextualizing the mobility changes in the context of number of cases would help us understand the impact of the interventions. I realize that measuring this impact is not the focus of the paper (which is fine) but again I think you have this information coming from the regression or could easily determine it with a slightly modified regression. Fourth, in the first regression model I see where absolute mobility enters into the analysis, i.e., via the strengths in the mobility network, but in the second analysis I agree that more motivation is needed for why relative change is more meaningful than absolute change. I understand why relative change is important, so I'm not suggesting abandoning that analysis, instead I think you could compare/contrast the absolute change (which will impact transmission). Finally, the reviewers also highlighted some confusion around Figure 1. In terms of additional historical context, I leave it up to the authors to decide whether that is needed.

I also think a recent paper on spatial scales of COVID-19 in Mexico is relevant to discuss, but note that I am an author on this paper and will not hold it against the authors if they disagree: https://academic.oup.com/pnasnexus/article/3/9/pgae306/7724517?login=true

We cannot make any decision about publication until we have seen the revised manuscript and your response to the reviewers' comments. Your revised manuscript is also likely to be sent to reviewers for further evaluation.

Sincerely,

Samuel V. Scarpino

Academic Editor

PLOS Computational Biology

Hannah Clapham

Section Editor

PLOS Computational Biology

I agree with the reviewers that this is an interesting paper demonstrating the importance of high-resolution mobility for measuring behavioral change during epidemics. I also agree with the reviewers that there are some aspects of the work which should be clarified and potentially some additional analysis are needed to support the conclusions (mostly depending on the resulting clarifications). For example, it's not clear how interventions were included in the regression model. Are they exclusively captured by in the mobility data and stratification of the analysis between intervention periods? If so, I might suggest a unified analysis with indicator variables associated with interventions (or continuous/ordinal). Second, the authors state that the network structures were strikingly similar. While I agree there are strong similarities, looking at the scatter plots of the Pijs there are many which have changed by 2+ orders of magnitude. I agree with the reviewers that more work is needed to articulate which aspects of the networks were retained and which were changed. I think you have this information in the regression analysis and do discuss it a bit, but I believe the work would be far strengthened with a deeper dive into differences/similarities in the networks. Third, I agree with the reviewers that contextualizing the mobility changes in the context of number of cases would help us understand the impact of the interventions. I realize that measuring this impact is not the focus of the paper (which is fine) but again I think you have this information coming from the regression or could easily determine it with a slightly modified regression. Fourth, in the first regression model I see where absolute mobility enters into the analysis, i.e., via the strengths in the mobility network, but in the second analysis I agree that more motivation is needed for why relative change is more meaningful than absolute change. I understand why relative change is important, so I'm not suggesting abandoning that analysis, instead I think you could compare/contrast the absolute change (which will impact transmission). Finally, the reviewers also highlighted some confusion around Figure 1. In terms of additional historical context, I leave it up to the authors to decide whether that is needed.

I also think a recent paper on spatial scales of COVID-19 in Mexico is relevant to discuss, but note that I am an author on this paper and will not hold it against the authors if they disagree: https://academic.oup.com/pnasnexus/article/3/9/pgae306/7724517?login=true

Reviewer's Responses to Questions

**Comments to the Authors:**

Reviewer #1: The authors here analyze whether historical mobile-phone data can be leveraged to predict how the implementation of non-pharmaceutical interventions during the COVID-19 pandemic shaped the mobility networks in two countries, Chile and Spain. Using a linear regression model with different socioeconomic, demographic and epidemiological predictors, the authors show how network metrics in a baseline scenario can shed light into the experience mobility reductions in the network. Likewise, the authors find that mobility network are resilient to the implementation of NPIs, both by comparing them directly with that corresponding to the baseline scenario or by simulating epidemics on top of them.

The article is generally well-written and easy to follow. While the scope of the manuscript is clear and the paper addresses a timely research question, in my opinion there are some flaws in the methodology that might substantially undermine the robustness and soundness of the results reported in the manuscript. In what follows, I would explain my concerns into detail, suggesting the authors how to address them.

One of the arguments used in the manuscript to support the resilience of the mobility network is that hotspots classification in terms of outbound mobility barely changes across the multiple waves of the disease. Nonetheless, in my opinion, raw outbound flows are not proper indicators to quantify such resilience as they are influenced by the differences in population among the different cities. Such differences could potentially turn population centers into mobility hotspots, regardless of how mobility was affected by NPIs. Hence, I think that computing hotspots computed according to per-capita out-strength are more suitable to showcase this consistence while removing the population bias.

Another important issue concerns the epidemic simulations carried out to support the similarity of epidemic trajectories corresponding to the mobility networks observed across multiple waves. The authors assume that the force of infection between two nodes just depends on the probabilities of travelling between these two nodes but not on the number total of individuals moving from each node. Neglecting the overall reduction in mobility of the different nodes could allow isolated nodes in the mobility network to propagate the disease across the population, therefore producing quite unrealistic epidemic trajectories. To solve this issue, I would encourage the authors to introduce the overall reduction in mobility of the different municipalities in their force of infection; their findings will hold as long as such reduction is roughly homogeneous across them.

As minor details:

- Figure 1: It is not clear to me what the colour code means. The authors should better explain it in the caption.

- Figure 3: I do not see the point of including Panels d and f as the spatio-temporal information of the similarity of epidemic trajectories is already shown by panels c and e. In addition, the authors should explain in the caption that the static networks corresponds to the average mobility networks observed in each period.

Reviewer #2: Although I know a lot about Chile, I don't have any expertise in the methods used in this manuscript. What strongly strikes me, though, is the utter lack of historical analysis. This is only the most recent of many epidemics in the country over the past centuries, from colonial times through the present. It is impossible to derive meaningful results without historical analysis. Simplistically using data from a single even without understanding it as part of a long-term historical process results in very little understanding of the basic processes involved. It will require major historical research to generate that sort of meaningful understanding of the epidemic.

Reviewer #3: Dear authors,

First of all, I would like to congratulate you on this insightful and valuable study. Your work provides important knowledge on the resilience of mobility networks during the COVID-19 pandemic and represents a solid contribution to pandemic preparedness. The combination of large-scale mobility data with spatial lag and regression models is particularly compelling, and your findings on the consistency of mobility routes across different intervention waves are both timely and relevant.

In addition to your valuable analysis, I would like to raise two suggestions for reflection and potential expansion in future discussions or conclusions of the article.

First, while your models predict certain routes and municipalities as critical in terms of mobility, it is not clear whether these predictions were compared with real-world outcomes during the COVID-19 pandemic in Chile. Specifically, were the routes identified by the model as the most critical also the ones most affected by the pandemic? And were the municipalities connected by these routes among those that experienced the greatest impacts? A comparison of these predictions with actual epidemiological data could strengthen the robustness of your findings and provide further confidence in the applicability of your model. I would appreciate it if you could clarify whether such a comparison was considered or whether there are plans to explore this in future work.

Second, while your study already highlights socioeconomic factors, I would encourage further reflection on how territorial and spatial characteristics of the municipalities could contribute to advancing the current model. For example, incorporating a spatial heterogeneity analysis that focuses on rural versus urban divisions, physical connectivity, or infrastructural limitations might reveal additional complexities in the mobility network's response to interventions. Understanding how these territorial dimensions interact with socioeconomic factors could offer new avenues for expanding the model and enhancing its applicability to different geographical contexts. This could be particularly valuable when considering public health strategies tailored to specific territorial realities.

I hope these reflections can contribute to further enhancing the impact of this important study.

Reviewer #4: The paper « Resilience of mobility network to dynamic population response across COVID-19 interventions: evidences from Chile » by Casaburi and colleagues studies how mobility data at the municipality-level along with other variables can be used to explain changes in mobility following interventions. Regression models were used to evaluate how sociodemographic, spatial, network and epidemiological factors impact the reduction in mobility after the first and second waves. They were also interested in how the mobility network was impacted by repeated interventions.

Major comment:

The paper was well written and the approach proposed by the authors was interesting. It was rigorous and I appreciated the thorough analysis of the mobility network.

However, I have a major concern. I am not a specialist in the use of mobility data but I am familiar with the evaluation of NPIs in epidemiology and I was surprised that spatial data on the NPIs were available but were not used in the model.

The regression model includes many covariates but the implementation of a NPI was not one of them. It seems to me that it would be the main driver of a reduction in mobility (that, and the implementation of NPI in the closest neighbors) and unless I missed something, it was not included in the model. I expect that including it as a covariate would lead to major changes in the model interpretation. To illustrate my point, age and gender were found to be significant factors for the change in mobility between the second and first wave. The authors explain this by stating that scarcely populated areas with a higher development index are mostly populated by young males. However, I am not convinced that those factors are drivers of the reduction in mobility. Rather, it could be that those specific regions with younger males were targeted by the NPIs during the second wave but not during the first wave.

Minor comment:

The map in Fig 1B is hard to read and would be better with four distinctive colors.

**Have the authors made all data and (if applicable) computational code underlying the findings in their manuscript fully available?**

Reviewer #1: **No: **Raw mobility data of the Chilean network is not made publicly availability due to privacy reasons.

Reviewer #2: None

Reviewer #3: Yes

Reviewer #4: Yes

PLOS authors have the option to publish the peer review history of their article (what does this mean?). If published, this will include your full peer review and any attached files.

Reviewer #1: No

Reviewer #2: No

Reviewer #3: **Yes: **Francesca FAGANDINI RUIZ

Reviewer #4: No
---

## [Decision Letter · Decision Letter 1]

16 Jan 2025

Dear Mr. Mazzoli,

We are pleased to inform you that your manuscript 'Resilience of mobility network to dynamic population response across COVID-19 interventions: evidences from Chile' has been provisionally accepted for publication in PLOS Computational Biology.

Best regards,

Samuel V. Scarpino

Academic Editor

PLOS Computational Biology

Hannah Clapham

Section Editor

PLOS Computational Biology

Reviewer's Responses to Questions

**Comments to the Authors:**

Reviewer #1: I would like to congratulate the authors for their very hard work to address all the concerns raised by both the referees and the editor. The new analyses included in the revised version of the manuscript firmly support the main message of the manuscript, i.e. the resilience of mobility networks across different epidemic waves and how informative past data can be for us to understand epidemic trajectories in future controlled scenarios.

I think this manuscript constitutes a highly valuable contribution for the literature. Therefore, I recommend its publication in PLOS Computational Biology.

Reviewer #3: Dear authors,

Thank you for your detailed responses to my comments. I appreciate the effort you have put into clarifying the scope of your study and addressing the limitations associated with data availability and methodological challenges. Your explanation of the focus on structural changes in the mobility network and the use of variables such as "urbanization," network metrics, and the IDC provide a clearer understanding of how territorial, spatial, and socioeconomic factors were incorporated into your analysis.

For my first comment, I acknowledge the difficulties in directly linking critical mobility routes to pandemic impacts due to data biases and the complexity of spatial transmission factors. Referring to related studies using genomic surveillance as an alternative approach adds significant value and situates your work within a broader research context. I commend your revisions to the Limitations Section and the addition of supplementary analyses, which enhance the manuscript’s transparency and rigor. Exploring these connections in future work, particularly with more reliable data sources, could provide valuable insights into the interaction between mobility patterns and epidemic outcomes.

Regarding my second comment, your clarification on the inclusion of spatial characteristics, such as the urban-rural divide, physical connectivity, and infrastructural limitations, strengthens the methodological framework of your study. The emphasis on the IDC and the effort to reference its composition explicitly in the manuscript enhance the robustness of your analysis. I encourage further exploration of how these territorial dimensions interact with socioeconomic factors, as this could offer a more nuanced understanding of mobility networks' responses to interventions, especially in geographically diverse contexts.

Overall, your thoughtful responses and the revisions you have implemented significantly improve the manuscript. I have no further comments and am confident that the changes made contribute meaningfully to the study's clarity and rigor. I encourage you to continue pursuing these lines of inquiry in future work to deepen the applicability of your findings to public health strategies in different territorial realities.

Reviewer #4: The comments were well addressed by the authors. I recommend accepting this paper.

**Have the authors made all data and (if applicable) computational code underlying the findings in their manuscript fully available?**

Reviewer #1: Yes

Reviewer #3: Yes

Reviewer #4: **No: **The authors provided detailed information on data availability but I couldn't find information on the code availability.

PLOS authors have the option to publish the peer review history of their article (what does this mean?). If published, this will include your full peer review and any attached files.

Reviewer #1: No

Reviewer #3: **Yes: **Francesca Fagandini

Reviewer #4: No